# Macroscopic Polarization Change via Electron Transfer in a Valence Tautomeric Cobalt Complex

Shu-Qi Wu [1], Meijiao Liu[1,2], Kaige Gao[1,3], Shinji Kanegawa[1✉], Yusuke Horie [4], Genki Aoyama[4], Hajime Okajima [4], Akira Sakamoto [4], Michael L. Baker [5,6], Myron S. Huzan [5,6], Peter Bencok[7], Tsukasa Abe [1], Yoshihito Shiota [1], Kazunari Yoshizawa[1], Wenhuang Xu[1], Hui-Zhong Kou [2] & Osamu Sato[1✉]

Polarization change induced by directional electron transfer attracts considerable attention owing to its fast switching rate and potential light control. Here, we investigate electronic pyroelectricity in the crystal of a mononuclear complex, [Co(phendiox)(*rac*-cth)] (ClO$_4$)·0.5EtOH (**1**·0.5EtOH, H$_2$phendiox = 9, 10-dihydroxyphenanthrene, *rac*-cth = racemic 5, 5, 7, 12, 12, 14-hexamethyl-1, 4, 8, 11-tetraazacyclotetradecane), which undergoes a two-step valence tautomerism (VT). Correspondingly, pyroelectric current exhibits double peaks in the same temperature domain with the polarization change consistent with the change in dipole moments during the VT process. Time-resolved Infrared (IR) spectroscopy shows that the photo-induced metastable state can be generated within 150 ps at 190 K. Such state can be trapped for tens of minutes at 7 K, showing that photo-induced polarization change can be realized in this system. These results directly demonstrate that a change in the molecular dipole moments induced by intramolecular electron transfer can introduce a macroscopic polarization change in VT compounds.

[1] Institute for Materials Chemistry and Engineering & IRCCS, Kyushu University, 744 Motooka, Nishi-ku, Fukuoka 819-0395, Japan. [2] Department of Chemistry, Tsinghua University, 100084 Beijing, PR China. [3] College of Physical Science and Technology, Yangzhou University, Jiangsu 225009, PR China. [4] Graduate School of Science and Engineering, Aoyama Gakuin University, 5-10-1 Fuchinobe, Chuo-ku, Sagamihara, Kanagawa 252-5258, Japan. [5] The Department of Chemistry, The University of Manchester, Manchester M13 9PL, UK. [6] The Department of Chemistry, The University of Manchester at Harwell, Didcot OX11 0FA, UK. [7] Science Division, Diamond Light Source, Didcot OX11 0DE, UK. ✉email: kanegawa@cm.kyushu-u.ac.jp; sato@cm.kyushu-u.ac.jp

Polarization, which characterizes the degree of co-alignment of permanent dipoles, is one of the most fundamental properties of materials[1]. When polarization is switched by the external stimuli, e.g. mechanical stress and temperature variations, temporary voltage and current are generated because of the change in the surface charge, which are known as piezoelectricity and pyroelectricity, respectively. Such phenomena are widely used in sensors, memory media, and capacitors[2–5]. Microscopically, polarization switching is mainly induced by collective ion displacement, molecular reorientation, and directional electron transfer[6,7]. The third mechanism has been increasingly applied in ferroelectrics in recent years[8–12], for it is expected that it exhibits higher switching rate and better durability compared with other switching mechanism of ferroelectric materials[13].

However, large polarization change can be realized in nonferroelectric molecular materials when the electron transfer direction of the target molecules in the crystal can be arranged in a noncancelable pattern, promising improved chemical design and control over the target systems. The change in their macroscopic polarization can also be detected by the generation of a pyroelectric current. In analogy, such phenomenon can be classified as electronic pyroelectricity, with inherent advantages of electronic ferroelectricity. To validate the applicability of this concept, we focus on a special family of coordination compounds that exhibit intramolecular electron transfer between metal centers and ligands, known as valence tautomeric (VT) complexes[14–34]. When crystallized in polar space group, the change in the dipole moments at molecular level should be amplified, and polarization change should be observable at single-crystal level. Furthermore, fast polarization change can potentially be induced by light-induced VT when a certain electronic absorption band is excited.

Herein, we report the detection of a pyroelectric current and photo-induced polarization change on the metastable states of [Co(phendiox)(*rac*-cth)](ClO$_4$)·0.5EtOH (**1**·0.5EtOH, H$_2$phendiox = 9, 10-dihydroxyphenanthrene, *rac*-cth = racemic 5, 5, 7, 12, 12, 14-hexamethyl-1, 4, 8, 11-tetraazacyclotetradecane) crystallized in the polar *P*2$_1$ space group as a proof-of-concept. As in the two-step VT process originated from the two enantiomeric complex motifs in different chemical environments[24], the pyroelectric current exhibits double peaks in the same temperature domain. Moreover, time-resolved Infrared (IR) spectroscopy shows that the metastable high-spin (HS) state can be generated by ultraviolet irradiation within 150 ps at 190 K, and such state is trapped for tens of minutes at 7 K, confirming the light-induced polarization change.

## Results

**Magnetic Properties**. Direct current magnetometry was performed to confirm the VT behavior (Fig. 1) of **1**·0.5EtOH with an external magnetic field of 2000 Oe. As shown in Fig. 2, in the first heating–cooling cycle between 5 and 300 K, temperature dependence of the susceptibility temperature products ($\chi_m T$, calculated

with respect to single cobalt ion) of **1**·0.5EtOH exhibited a reversible behavior. However, in the second measurement cycle up to 400 K, distinct temperature dependence was observed, and the transition temperature in cooling process, which was measured after the heating process, shifted to a lower temperature. On the other hand, in the third measurement cycle, the $\chi_m T$ values approximately coincided with those recorded in the cooling process in the second cycle without hysteresis loop, indicating the presence of a solvent effect (Supplementary Fig. 1). Solvent loss at high temperatures led to a decrease in the transition temperatures, which could possibly be explained by stabilization of the HS species with reduction of the steric hindrance. The $\chi_m T$ values nearly vanish below 180 K, consistent with the spin-singlet ground state. The $\chi_m T$ value of 3.0 cm$^3$ K mol$^{-1}$ at 400 K is close to that expected for a HS Co$^{II}$ ion ($S = 3/2$) and a semiquinonato-form ligand ((phenSq)$^-$) with a single unpaired electron. To obtain more information about the transition behavior, we analyzed the first-order derivatives of the $\chi_m T$ values with respect to temperature (Fig. 2, inset). Two maxima were found and fitted by the sum of two Gaussian functions, which demonstrated the existence of two distinct magnetic transition processes centered at 251 and 333 K in the heating process. The two maxima of the first-order derivative remained in the cooling process with the transition temperatures of 240 and 312 K, which were different from those obtained during the heating process[24]. The change in the magnetic moment corresponds to the two distinct VT processes of **1**·0.5EtOH. We also investigated a reversible temperature range and found that the $\chi_m T$ values of solvated samples could repeatedly be observed without significant change when measured below 330 K to prevent solvent loss (Supplementary Fig. 2).

**Crystal Structures and Calculations**. Single-crystal structures of **1**·0.5EtOH at 123 K (low-temperature phase), 293 K (intermediate-temperature phase), and 393 K (high-temperature phase) were carefully analyzed to characterize the structural change during the electron transfer process (Fig. 3 and Supplementary Fig. 3). Tables S1 and S2 summarize the crystal data, the structure refinement parameters and the selected bond lengths. Unlike the reported structures of [Co(phendiox)(*rac*-cth)](PF$_6$)·1.5CH$_2$Cl$_2$ crystallized in the achiral $C^2/c$ space group[23], **1**·0.5EtOH crystalizes in the polar *P*2$_1$ space group possibly driven by the weak hydrogen bonds between perchlorate anion and cth ligands

## Fig. 1

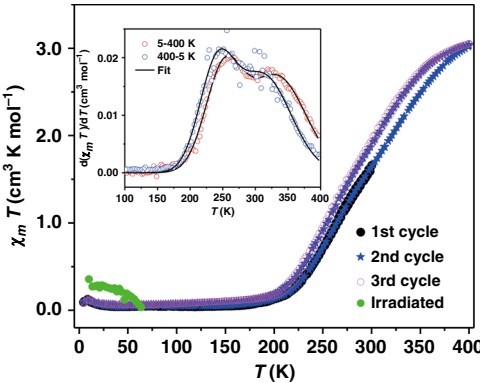

**Fig. 2 Magnetic properties of 1·0.5EtOH.** Temperature-dependent $\chi_m T$ product and light-induced valence tautomerism effect induced by laser. The measurement was performed with a sweeping rate of 5 K min$^{-1}$ in a 5–300–5 (first cycle)–400–5 (second cycle)–400–5 K (third cycle) sequence. Inset: first-order derivatives of the $\chi_m T$ product and double Gaussian fitting (black lines) in the second measurement cycle (5–400–5 K).

**Fig. 1 VT in 1·0.5EtOH.** Schematic representation of the electron transfer behavior of the complex motif.

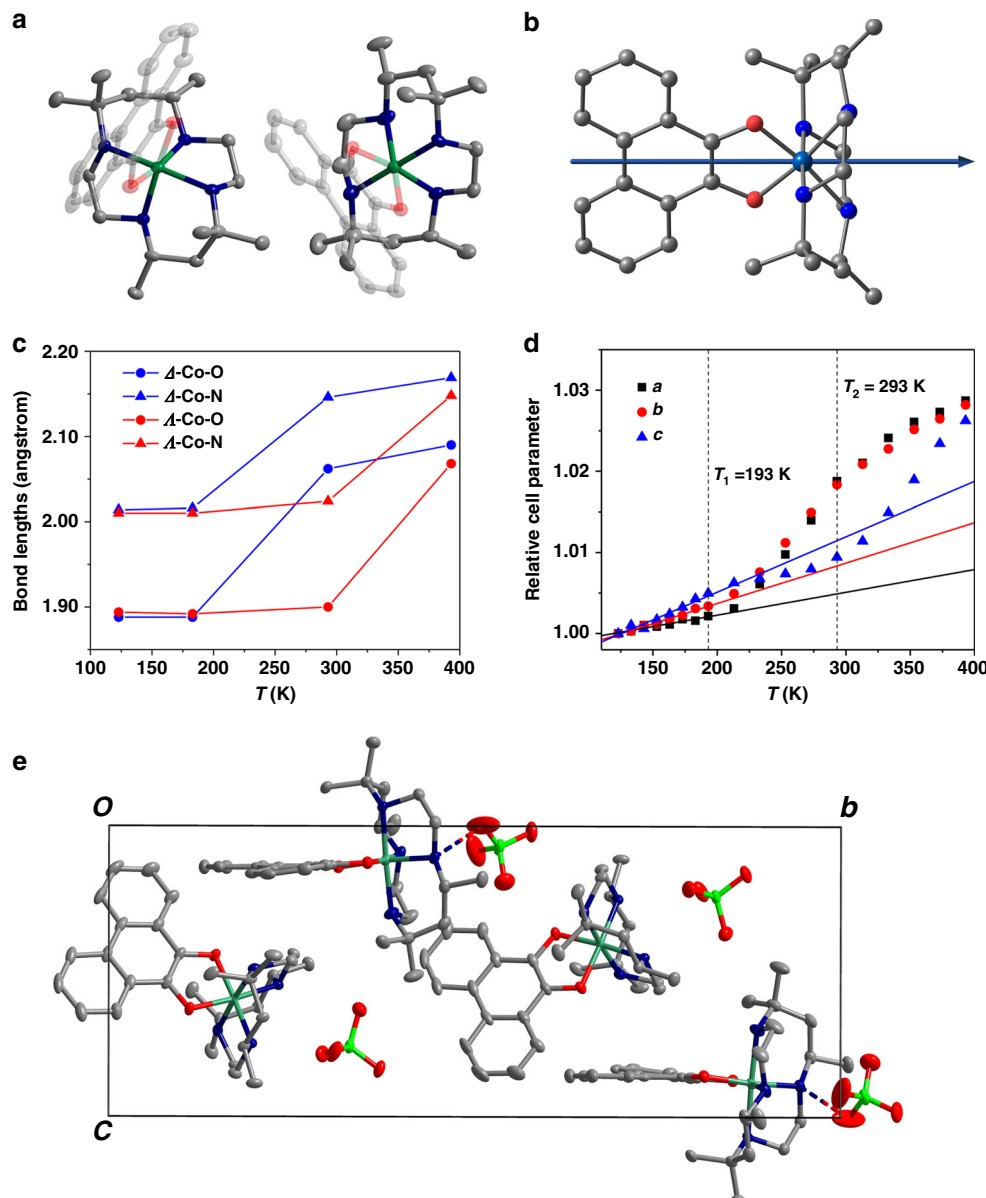

**Fig. 3 Crystal structures and packing modes of 1·0.5EtOH. a** Crystal structure of the pair of $\Delta$-[Co(phendiox)(cth)] (left) and $\Lambda$-[Co(phendiox)(cth)] (right) in **1**·0.5EtOH at 123 K. Hydrogen atoms, anions, and solvent molecules are omitted for clarity; **b** calculated direction of the molecular dipole moment; **c** temperature-dependent bond lengths around Co centers; **d** temperature-dependent cell parameters normalized with respect to their values at 123 K. Solid lines represent the linear fitting of the thermal expansion parameters; **e** crystal packing viewed along the *a*-axis. The *b*-axis is the polar axis. Cobalt is shown in malachite, oxygen in red, carbon in gray, nitrogen in navy, and chlorine in green. Hydrogen atoms and solvent molecules are omitted for clarity. Hydrogen bonds between perchlorate groups and the $\Lambda$-Co motifs are represented by dashed lines.

(Supplementary Fig. 4). The asymmetric unit is composed of an enantiomeric pair of cobalt motifs ($\Lambda$-[Co(phendiox)($RR$-cth)]$^+$ and $\Delta$-[Co(phendiox)($SS$-cth)]$^+$), two perchlorate anions, and one ethanol molecule (Fig. 3a and Supplementary Fig. 5). The Co ion is 6-coordinated by four N atoms from the cth ligand and two O atoms from the phendiox ligand. At 123 K, the Co–O and Co–N bond lengths range from 1.887(2) to 1.897(3) Å and from 1.997(3) to 2.031(3) Å, respectively, while the C–O bond lengths range from 1.348(4) to 1.358(4) Å. These bond lengths are typical of catecholato-form phendiox ligand (($phenCat)^{2-}$) and the low-spin (LS) Co(III) ion. At 293 K, the coordination bond around the $\Delta$-Co motif exhibits a significant elongation with Co–O and Co–N bond lengths ranging from 2.058(3) to 2.065(3) Å and from 2.127(3) to 2.175(3) Å, respectively. The C–O bond lengths shrink correspondingly to 1.289(5) and 1.297(5) Å. Such changes

are consistent with the typical VT behavior of the cobalt–dioxolene family[20]. However, the coordination bond lengths in the $\Lambda$-Co motif only exhibit very slight changes (Fig. 3c and Supplementary Table 2). Structural analysis reveals a weak hydrogen bond in the $\Lambda$-Co motif and the perchlorate group with a N–H---O distance of 2.976(9) Å at 123 K, whereas the weak contacts around the $\Delta$-Co motif have distances >3.078 Å. The weaker interaction between perchlorate anion and the $\Delta$-Co motif was also confirmed by the calculated electronic densities (Supplementary Fig. 6). The difference in the local steric hindrance may explain the different VT behavior of the two motifs[23]. The observed solvent effect by magnetometry also supports this explanation: when the solvent molecules in voids are removed from the lattice, steric hindrance is reduced, leading to a decrease in the transition temperatures. It should be noted that the cobalt

motif undergoing electron transfer is solely determined by its interaction with perchlorate anions and not by its absolute structure. In enantiomeric crystals, the $\Lambda$-Co motif exhibits electron transfer first. Upon further heating to 393 K, both motifs show characteristic bond length of the $[Co^{II}(phenSq)(cth)]^+$ form Co–O and Co–N bond lengths ranging from 2.064(4) to 2.098(3) Å and from 2.108(5) to 2.198(4) Å, respectively, and C–O bond lengths ranging from 1.277(6) to 1.298(6) Å. These observations clearly reflect the two-step VT process from the two cobalt enantiomers in **1**·0.5EtOH (Supplementary Fig. 7).

Notably, the $\Lambda$-[Co(phendiox)(RR-cth)]^+$ and $\Delta$-[Co(phendiox)(SS-cth)]^+$ motifs are oriented in similar directions in the lattice with angles of 27.4° and 25.1° between the O–Co–O bisection line and the crystalline $b$-axis at 123 K, respectively. The molecular pairs are further extended by the $2_1$-screw symmetry, creating a directional electron-transfer vector within the crystal (Fig. 3e). Upon heating, the projection angles only change very slightly to 28.6° and 25.4° at 293 K, and 28.1° and 25.1° at 393 K. Hence, the change in the electronic structures induced by electron transfer between the ligand and the Co ion will change molecular dipole moments and the polarization along the crystalline $b$-axis at the single-crystal level.

The change in the electronic structures and dipole moments of **1**·0.5EtOH is further confirmed by the density functional theory (DFT) calculations. In the closed-shell singlet state, $[Co^{III}(phenCat)(cth)]^+$ possesses a permanent electric dipole moment of 13.15 Debye, and is oriented along the direction from the ligand to the Co ion, coincident with the pseudo-$C_2$ axis of the molecule (Fig. 3b). In the quintet state, the spin density is mainly distributed on the cobalt center and the two oxygen atoms from the phendiox ligand (Supplementary Fig. 8), confirming the electronic structure of $[Co^{II}(phenSq)(cth)]^+$. The calculated electric dipole moment is 7.27 Debye, without significant change of its direction compared with that of the closed-shell singlet state. The calculation of the permanent dipole moment as the linear coefficient of the first-order Stark effect gives similar values of 13.39 Debye for the closed-shell singlet state and 7.52 Debye for the quintet state, respectively (Supplementary Fig. 9)[25]. The decrease in the dipole moment of the quintet state is attributed to the electron transfer from $(phenCat)^{2-}$ to the $Co^{III}$ ion, confirming the electron transfer in the **1**·0.5EtOH.

**Pyroelectric Measurements**. Magnetometry and single-crystal analysis confirm that a two-step electron transfer occurs on the molecular level, and the large change in the molecular dipole moments before and after electron transfer leads to the thermal polarization change. Notably, **1**·0.5EtOH crystallizes in the polar $P2_1$ space group with approximately the same projection angle between the electric dipole orientation and the polar $b$-axis. When projected to the crystalline $b$-axis, the change in the magnitude of the net dipole moments per unit cell can be estimated at each temperature by $\Delta P = [\Delta(\mu_{Co1}\cos\theta_1) + \Delta(\mu_{Co2}\cos\theta_2)]/V_{cell}$, where $V_{cell}$ is the volume of the unit cell, and $\mu_{Coi}$ and $\theta_i$ refer to the magnitude of dipole moment and the angle between dipole moment vector and crystalline $b$-axis of the $i$th molecular motif, respectively (Supplementary Table 3). Such values could be compared with the experimental results.

To evaluate the change in macroscopic polarization during the electron transfer process, the pyroelectric current was measured on a single crystal with a Keithley 6517B electrometer and the MPMS chamber working as a temperature controller (see "Methods"). Measurements were performed between 100 and 330 K where the solvent effect is negligible as mentioned above. With changing temperatures, two broad peaks of the pyroelectric coefficient were clearly observed in the similar temperature range

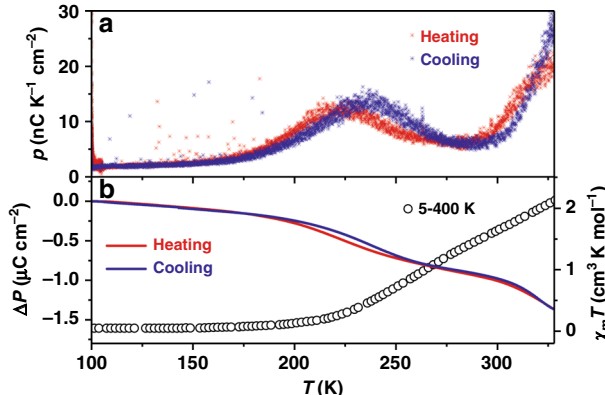

**Fig. 4 Pyroelectric properties and polarization change between 100 and 330 K. a** Pyroelectric coefficient ($p$, red and blue stars) on a single crystal of **1**·0.5EtOH between 100 and 330 K with a sweep rate of 5 K min⁻¹; **b** polarization change ($\Delta P$, red and blue lines) as comparison with the $\chi_m T$ products (open circles).

where the two electron transfer processes occur as shown by the single-crystal analysis and magnetometry (Fig. 4a). A different temperature sweep rate gave nearly identical pyroelectric coefficient, confirming that the observed current mainly came from pyroelectricity instead of the current leakage (Supplementary Figs. 10 and 11).

Integration of current over the time of interest yields the change in macroscopic polarization (Fig. 4b). In the temperature range between 123 and 183 K (before electron transfer), the integrated change is *ca.* 0.14 μC cm⁻². The value is much larger than 0.01 μC cm⁻² that was estimated theoretically based on the electron transfer and is ascribed to the secondary pyroelectric effect, originating from the piezoelectric effect during the thermal expansion of polar crystals[1]. To test this assumption, the thermal expansion parameters were examined. Below 183 K, when the electron transfer process has not occurred, the lattice exhibits a positive thermal expansion with linear expansion coefficients $\alpha_a$, $\alpha_b$, and $\alpha_c$ of $28(4) \times 10^{-6}$, $50(3) \times 10^{-6}$, and $68(8) \times 10^{-6}$ K⁻¹ and a volumetric expansion coefficient $\alpha_v$ of $145(6) \times 10^{-6}$ K⁻¹ (Fig. 3d and Supplementary Fig. 12). These values are located between those of ferroelectric inorganics, which are typically below $10 \times 10^{-6}$ K⁻¹ ($PbZr_{1-x}Ti_xO_3$ (PZT)[35], $5.4 \times 10^{-6}$ K⁻¹; wurtzite ZnO[36], $6.5 \times 10^{-6}$ K⁻¹), and ferroelectric polymers, which are typically above $100 \times 10^{-6}$ K⁻¹ (polyvinylidene difluoride (PVDF), 120–145 × 10⁻⁶ K⁻¹)[37]. Detailed calculation of the secondary pyroelectric coefficient requires other knowledge of the mechanical and piezoelectric parameters. However, the low-temperature pyroelectric coefficient of **1**·0.5EtOH is higher than those of PZT and ZnO and lower than that of PVDF, within a reasonable range. Notably, the pyroelectric coefficient remains *ca.* 1.9 nC K⁻¹ cm⁻² at 100 K when no electron transfer process occurs, comparable to that of the PVDF film (2.7 nC K⁻¹ cm⁻²)[37].

The observed change in polarization between 183 and 293 K (during the first electron transfer process) is *ca.* 0.77 μC cm⁻², slightly higher than the theoretically estimated 0.63 μC cm⁻². Considering the secondary pyroelectric contribution, the result satisfactorily confirms that the polarization change mainly originates from the electron transfer process. Therefore, the phenomenon should be appropriately termed "electronic pyroelectricity." It should be noted that the calculated polarization is derived purely from the intramolecular electron transfer process. Other origins of dipole moments in the lattice, e.g. the crystalline ethanol molecules and anions (Supplementary Table 4), are expected to contribute to lesser degrees to the polarization

change. On the other hand, a significant discrepancy in the peak temperatures obtained from the pyroelectric current (236 K in the cooling process and 224 K in the heating process) and from the first-order derivatives of $\chi_m T$ values (251 K) was also observed. The lowering of transition temperature in the pyroelectric measurement can be explained by the difference between these two techniques: magnetometry refers to the bulk of the crystalline sample, whereas pyroelectricity is more sensitive to the surface layer. The molecules at the surface suffer less chemical pressure, and may possibly exhibit a lower transition temperature. Indeed, when probed by X-ray absorption spectroscopy with the total electron yield detection mode that is extremely sensitive to the surface layer, a much higher ratio of HS molecules was detected compared with that obtained by magnetometry in the reported VT cobalt–dioxolene complexes[38–40]. However, the preliminary experiments showed that **1**·0.5EtOH was very sensitive to X-ray exposure and exhibited irreversible X-ray-induced damage that hindered further investigations of the surface effect using this technique. The discrepancy in the peak temperatures in the heating and cooling processes requires more careful studies. Upon further heating above 293 K, the pyroelectric current increases again, showing a second peak, due to the occurrence of the second electron transfer process. However, the measurements could not be extended to the temperature (*ca.* 400 K), when the electron transfer process finishes to estimate the polarization change, because desolvation introduces defects in the crystalline sample, leading to increasing leakage current. The thermal movements and partial loss of ethanol molecules also complicate the theoretical calculation of the polarization change. Notably, during the electron transfer process, the thermal expansion of the lattice is significantly enhanced by the increase of the molecular size and deviates from the linear behavior. The positive thermal expansion of **1**·0.5EtOH is expected to further enhance the pyroelectric property because the molecular dipole moments decrease during the electron transfer process.

To conclude, two pyroelectric current peaks are detected in the same temperature domain where the two-step electron transfer occurs, confirming the electronic pyroelectricity behavior of single crystal of **1**·0.5EtOH.

**Photo-induced IR Spectroscopy and Magnetometry.** Advantageous over ion displacement, electron transfer can be directly controlled by light, providing the possibility of light-induced polarization change. This is a known effect, used in pyroelectric laser sensors. However, the mechanism is completely different in this case. In pyroelectric laser sensors, light works as a heat source[29], whereas, in our situation, light directly excites the molecule and drives the change in its electronic structure, hence the change in macroscopic polarization[18]. Here we performed temperature-dependent, photo-induced, and time-resolved IR spectroscopy and photomagnetometry to evidence the light-induced polarization change.

To investigate the dynamics from the ground to the light-excited metastable states, the time-resolved IR absorption measurements were performed at 190 K using femtosecond laser pulses with a transmitting geometry, adopting the pump-probe technique[26,30–34]. The picosecond timescale evolution of transient IR ($\Delta A$) spectra with a 380-nm pump pulse was plotted in Fig. 5. Similar spectral features with the electron-transferred state, [Co$^{II}$(phenSq)(cth)]$^+$, were observed with the picosecond time-resolved measurements; the bands at 1365 and 1374 cm$^{-1}$ decrease and the band at 1476 cm$^{-1}$ increases. The transient IR spectra at longer delay times (over 500 ps) were consistent with the temperature-induced IR difference spectrum (Fig. 5a and Supplementary Fig. 13). Apart from these spectral features, two

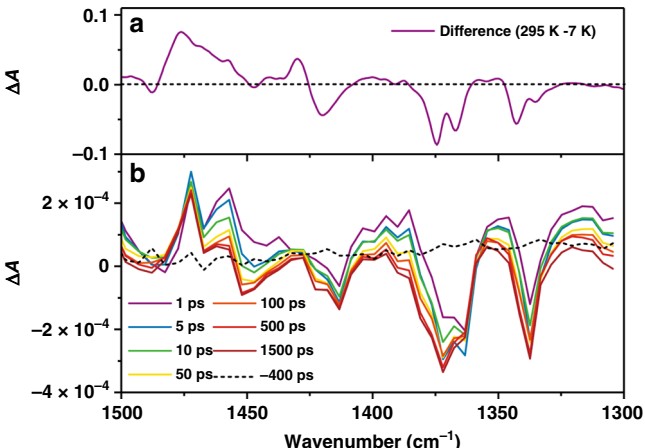

**Fig. 5 Time-resolved transient IR difference spectra and the temperature-induced spectrum. a** Temperature-induced IR difference spectrum between 295 K and 7 K; **b** picosecond timescale evolution of transient IR difference spectra with a 380-nm pump pulse at 190 K in the first 1500 ps.

new peaks transiently appear in the first several tens of picoseconds at 1457 and 1386 cm$^{-1}$, and then gradually declined. These features were tentatively assigned to the light-induced intermediates. The time evolution of $\Delta A$ was fitted by a double exponential function, and similar relaxation times were found within 150 ps for both absorptions at 1457 and 1386 cm$^{-1}$ (Supplementary Fig. 14), consistent with the two-step inter-conversion behavior observed in the same family of compounds[30]. After the intermediates disappear, the transient IR spectra do not change at all, indicating that the electron transfer has finished within 150 ps. As a result, the fast light-induced electron transfer process, and hence ultrafast polarization change is realized in this system.

To confirm the possibility to trap the polarization-changed metastable state at lower temperature, IR spectra of the powder sample attached to a CaF$_2$ plate were recorded with irradiating a 365-nm emission light from a high-pressure mercury lamp onto the sample surface to excite the $\pi$–$\pi^*$ transition of the catechol-form ligand (Supplementary Figs. 15–17). During the UV irradiation at 7 K, the IR spectrum became closer to that observed at 295 K without irradiation (Supplementary Figs. 13 and 18), suggesting the presence of trapped electron-transferred metastable species, [Co$^{II}$(phenSq)(cth)]$^+$. Photomagnetic measurements were also performed. The sample was irradiated with a blue laser with a central wavelength of 405 nm for 2 h, resulting in photoconversion of 7%. The trapped metastable HS species relaxed back with temperature increase and time evolution. In addition, no HS species could be detected by magnetometry at temperatures above 62 K (Fig. 2). To estimate the dynamics of the relaxation process, the time-dependent decay of magnetization was recorded at 7 K, and was fitted with the stretched exponential law, giving a relaxation time of 1158 (24) s with a time distribution parameter of 0.74(1). The distribution in the relaxation time indicates that a certain amount of molecules relaxes to the ground state in a very short time, which may partially explain the low photoconversion observed in our system (Supplementary Fig. 19)[28]. However, the long-lived metastable molecules still contribute to the observed light-induced VT, consistent with the results obtained from the IR spectra. Therefore, the polarization-changed metastable state via light-induced electron transfer can be trapped for tens of minutes at 7 K.

In conclusion, we characterized the electronic pyroelectric behavior of a mononuclear cobalt complex, **1**·0.5EtOH. It exhibits a two-step thermally induced VT behavior, as demonstrated by temperature-dependent IR spectra, single-crystal analysis, and magnetic measurements. DFT calculations revealed a significant change in the molecular dipole moments for the HS and LS states. Its polar crystalline space group enables the direct observation of the accumulative change of the dipole moments on the single-crystal level, leading to the observation of a pyroelectric current during the electron transfer process in non-ferroelectric molecular compounds. Moreover, time-resolved IR spectroscopy showed that the metastable HS state could be generated by ultraviolet light irradiation within 150 ps at 190 K, confirming the ultrafast light-induced polarization change. The metastable state was trapped and exhibited a lifetime over tens of minutes at 7 K. Notably, the electronic pyroelectricity characterized here is fundamentally different from common ferroelectricity; no electric field is needed to preliminarily polarize the system to optimize the pyroelectric coefficient, which is appropriate for pyroelectric current based applications and more energy economic.

## Methods

**Synthesis**. All solvents and reagents were used as received from Tokyo Chemistry Industry Co, Ltd.

The racemic cth ligand was prepared according to the literature[41]. A total of 111 g of 60% perchloric acid (0.66 mol) was slowly added into the acetone solution (1 L) of 1,2-diaminoethane (40 g, 0.67 mol) in an ice bath and was stirred overnight. The white solid of 5, 7, 7,12, 14, 14-hexamethyl-1, 4, 8, 11-tetraazacyclotetradeca-4, 11-diene diperchlorate was collected by filtration and was washed with acetone without further purification (126 g, 39.7%). Then, the white crude product (100 g, 0.21 mol) was dissolved in methanol (500 mL) in an ice bath. The solid mixture of $NaBH_4$ (19 g, 0.63 mol) and NaOH (16.5 g, 0.42 mol) was slowly added in portions. After the solution cooled down to room temperature, an aqueous solution of NaOH (50 g in 1 L water) was added, and the solution was stirred for another 1 h. White solid was obtained by filtration and dried in air overnight. Then, the product was dissolved in 600 mL of methanol. Water (400 mL) was added into the methanol solution and stirred for 1 h; the precipitate of *meso*-cth was formed and was discarded by filtration. Another 200 mL of water was added to the solution, and the precipitate formed after stirring for 1 h was discarded again by filtration. The solution was evaporated to near dryness on a rotatory evaporator, and the white solid product of racemic cth monohydrate was collected by filtration and washed with a small amount of cold water (15.2 g, yield 23.9%).

$H_2$phendiox was obtained by the reduction of 9, 10-phenanthrehequinone. The latter compound (2.08 g,10 mmol) was dissolved in methanol (10 mL). Solid $NaBH_4$ was slowly added into the solution in portions until the solution became colorless. Then, hydrochloric acid (10%, 50 mL) was added, and white flocculent precipitate was formed. After cooling down to room temperature, the product was collected by filtration (1.42 g, yield 66.9%).

$[Co^{II}(OAc)](rac$-cth$)](ClO_4)$ was synthesized according to literature with slight modification[22]. A mixture of $Co(AcO)_2·4H_2O$ (3.0 g, 12 mmol, AcO = acetate) and racemic cth (3.4 g, 12 mmol) in EtOH (20 mL) was heated to 60 °C under $N_2$ atmosphere. After dissolving, solid $NaClO_4$ (2.5 g, 20 mmol) was added. $[Co^{II}(OAc)(rac$-cth$)](ClO_4)$ gradually formed as a pink solid. After the reaction mixture was cooled in an ice bath, the pink precipitate was collected by filtration and washed with cold EtOH followed by $Et_2O$ (3.3 g, yield 54.8%).

An equimolar amount of $AgClO_4$ (414 mg, 2.0 mmol) in $H_2O$ (5 mL) was added to the solution of $[Co^{II}(OAc)(rac$-cth$)](ClO_4)$ (1.00 g, 2.0 mmol) in acetone (20 mL). After stirring at room temperature for 30 min, the precipitated inorganic solid was removed by filtration. The solvent was removed under reduced pressure. $[Co^{III}(OAc)(rac$-cth$)](ClO_4)_2$ was obtained as a purple solid (0.93 g, yield 77.5%). Then, a mixture of the resulting $[Co^{III}(OAc)(rac$-cth$)](ClO_4)_2$ (601 mg, 1.0 mmol) with $H_2$phendiox (210 mg, 1.0 mmol) and NaOH (80 mg, 2.0 mmol) in MeOH (25 mL) was heated to reflux under $N_2$ atmosphere. After stirring for 10 min, the solvent volume was reduced to 5 mL with the flow of $N_2$ gas. After the reaction mixture was cooled in an ice bath, the precipitate was collected by filtration and washed with a small amount of cold 2-propanol, followed by pentane. **1**·0.5EtOH was obtained in the form of dark-colored crystals via recrystallization from hot ethanol (402 mg, yield 61.7%).

**X-ray Structure Determination**. All single crystals were coated with an oil-based cryoprotectant and mounted on nylon loops. Diffraction data were collected at 123, 183, 293, 393 K and again at 123 K with the same single crystal of **1**·0.5EtOH under a cold nitrogen gas stream on a Rigaku FR-E+ diffractometer equipped with a HyPix-6000 area detector, using multi-layer mirror monochromated Mo–$K\alpha$

radiation ($\lambda = 0.71073$ Å). The structures were solved by a direct method and refined via full-matrix least-squares on $F^2$ using the SHELX program[42,43] implemented in the OLEX2 program[44] with anisotropic thermal parameters for all non-hydrogen atoms. The hydrogen atoms were geometrically added and refined by the riding modelRoom-temperature powder, X-ray diffraction patterns were recorded on a Rigaku-TTR diffractometer to examine crystal purity (Supplementary Fig. 20).

**IR Spectroscopy**. Temperature-dependent IR spectra were recorded by using an FT-IR spectrophotometer (VERTEX 70, Bruker) equipped with a closed-cycle helium refrigerator cryostat (Nagase Techno-Engineering). The ground-powdered samples were held between a grained and a plane $CaF_2$ plates. To excite the sample, the 365-nm emission line from a high-pressure mercury lamp (Optical Modulex, USHIO) was used. The photo-induced IR spectra were measured with irradiating UV light onto the sample for 30 min.

The experimental setup of the picosecond time-resolved IR absorption measurements is described as follows. The fundamental output from a Ti:sapphire regenerative amplifier (Solstice Ace, Spectra Physics; wavelength 800 nm, power 4.5 W, repetition rate 1 kHz, pulse width ~100 fs) was divided into two beams. Both were used to excite two optical parametric amplifiers (OPAs) and generate pump and probe pulses. The probe IR pulse was obtained by difference-frequency generation between the signal and idler waves from one OPA (TOPAS-C, Light Conversion). The center wavelength of the IR pulse was 6700 nm, and its spectral bandwidth was ~170 cm$^{-1}$ in FWHM. The probe IR beam was divided into two paths with a ZnSe half mirror. One portion of the beam was focused on and transmitted through the sample; it was used as a sample beam. The other was used as a reference beam to cancel out intensity fluctuation of the IR pulses. Both were introduced into a 19 cm spectrograph (TRIAX190, HORIBA JOBIN YVON) with a slightly different height offset. The two dispersed beams were simultaneously detected by a $2 \times 64$-channel liquid-nitrogen-cooled HgCdTe detector array and integrated by 128 box-car integrators (IR-12-128, InfraRed Associates). The normalized IR signals were obtained by dividing the sample intensities by the reference intensities. The pump ultraviolet pulse was centered at 380 nm. The pump beam was modulated at half the repetition rate of the probe beam (500 Hz) by a mechanical chopper. The modulated pump beam was passed through a delay stage and focused on the sample noncollinearly relative to the probe beam. The normalized IR signals with and without the pump pulses were separately accumulated by a computer. The pump-induced IR absorption was obtained by dividing the pump-on signal by the pump-off signal. The cross-correlation time between the pump and probe pulses, which was determined by the rise of a transient IR absorption of photoexcited silicon due to free carriers, was ~0.5 ps. The energy of the probe pulse at the sample position was <2 μJ, and that of the pump pulse was <1 μJ. **1**·0.5EtOH was held between a grained and a plane $CaF_2$ plates. The sample was placed in a liquid-nitrogen-cooled cryostat (OptistatDN-V, Oxford) and kept at 190 K where most of the sample was in the LS state. The total exposure time for each time delay was 15 min.

**Other Physical Measurements**. Thermogravimetric analysis was performed on a DTU-2A equipment at 293–473 K with a heating rate of 10 K min$^{-1}$ in an air atmosphere. A polycrystalline sample was used, and the gradual weight loss was found in the whole temperature range due to the solvent loss.

Temperature-dependent UV–vis absorption spectra were obtained using a UV-3100PC (Shimadzu) scanning spectrophotometer with a helium-flow-type refrigerator. Powdered samples were attached to the transparent tape.

Magnetic susceptibility measurements were conducted on a Quantum Design MPMS-XL superconducting quantum interference device magnetometer under a 2000 Oe field, with a sweeping rate of 5 K min$^{-1}$ in a 5–300–5–400–5–400–5 K sequence. The measurement samples were prepared by encapsulating microcrystal sample of **1**·0.5EtOH into a gelatin capsule. Photomagnetic measurements were performed on the powdered sample attached to transparent tape. A MDL405-100-mW violet laser was adopted as the excitation source. The sample was irradiated for 2 h with the cooling valve open to maintain a low-temperature environment. The relaxation measurements were performed at 7 and 20 K.

Pyroelectric measurements were performed with Keithley 6517B electrometer and the Quantum Design MPMS-XL chamber as temperature controller. The single-crystal sample ($1 \times 0.35 \times 0.15$ mm$^3$) was sandwiched between the silver pastes on its (010) and (0–10) surfaces (Supplementary Fig. 21). No electric field was applied to the sample. The measurement temperature was restricted between 100 and 330 K under helium gas flow. Magnetic measurements were also performed under the same conditions to show that the transition behavior was reversible in this temperature range, and no significant solvent effect was found. To ensure the reliability of the current recorded, measurements were performed when the background current was below 0.02 pA (Supplementary Fig. 22). The measurements were conducted with the temperature sweep rate of 5 and 10 K min$^{-1}$, respectively (Fig. 4 and Supplementary Fig. 10). The temperature range was extended to 350 K where the desolvation was not significant in the homemade aluminum shield box. Similar temperature dependence of pyroelectric coefficient was observed (Supplementary Fig. 11). However, a leakage current was detected when the measurements were extended to a higher temperature possibly due to defects generated from desolvation.

**DFT Calculations**. All calculations for the complex **1** were carried out with the Gaussian 09 package[45]. The B3LYP* functional[46] used for all energy calculations is a reparametrized version of the B3LYP functional[47,48] with 15% Hartree–Fock exchange, which is expected to provide better performance for accurate spin-state splitting, whereas the original B3LYP functional would result in overestimation of the HS stability. For the Co atoms, the (14s9p5d)/[9s5p3d] primitive set of Wachters–Hay[49,50] with one polarization *f*-function was used, and for the H, C, N, and O atoms, the 6-311+G** basis set[51] was also used. Molecular structures were adopted as the initial guess. The geometry optimizations in quintet and triplet states were carried out separately. The quintet and triplet states were optimized separately for the HS state. The quintet state was found to be more stable, corresponding to the ferromagnetically interacted ground state. It should be noted that more sophisticated post-Hartree–Fock calculations provide the spin-orbit coupled admixture of quintet and triplet configurations as a better description of the ground electronic configuration for the Co-semiquinonato ligand. However, the dipole moment calculated by CASSCF/NEVPT2 protocol using ORCA 4.1.2 package with def2-TZVP basis is 6.99 Debye[52–54], which is similar to that obtained from the calculation of the quintet state by DFT method[55]. The molecular structures of the singlet and quintet states were also optimized with the B3LYP functional, which provided similar results. After geometry optimizations, vibrational analyses were performed to ensure that no imaginary frequencies were found, and the calculated IR spectra based on the frequency analysis were in good agreement with the experimental ones, demonstrating the reliability of the optimized geometries. To compare intermolecular interactions between Co-centered motifs and perchlorate anions at 123 K, a restricted optimization was also performed by relaxing the hydrogen atoms while freezing all the other atoms. All the optimized coordinates are summarized in Supplementary Tables 5 and 6. TD-DFT calculations were used to calculate all the vertical spin-allowed electronic transitions at the same level of theory with geometric optimizations[56,57]. The 60 excited states were considered, covering the bands from 250 to 800 nm. The charge density difference was analyzed by Multiwfn 3.7 package[58].

## Data Availability

The X-ray crystallographic data for the structures at different temperatures reported in this study have been deposited at the Cambridge Crystallographic Data Centre (CCDC), under deposition numbers CCDC1952512-1952516. These data can be obtained free of charge via http://www.ccdc.cam.ac.uk/conts/retrieving.html or from CCDC (12 Union Road, Cambridge CB2 1EZ, UK; Fax: +44 1223 336033; e-mail: deposit@ccdc.cam.ac.uk). Additional data are available from the authors upon request.

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

## Acknowledgements

This work was supported by MEXT KAKENHI (Grant Numbers 20H00385, 17H01197, 17K05761, and 16K05725), National Natural Science Foundation of China (Grant Number 21971142), Natural Science Foundation of Jiangsu Province (Grant Number BK20170482). XAS experiments were carried out with the support of the Diamond Light Source (proposal SI21117). The synchrotron radiation experiments were performed at the BL02B1 of SPring-8 with the approval of the Japan Synchrotron Radiation Research Institute (JASRI) (Proposal Nos. 2019B1272 and 2018B1259). S.-Q.W. thanks Dr Hiroyasu Sato from Rigaku Corporation for assistance in single-crystal analysis. Mr Takeo Ishikawa is acknowledged for finding the solvatomorphism in this system. M.L. and K.G. thank the China Scholarship Council for support.

## Author Contributions

S.K. and O.S. supervised the study. S.-Q.W. performed the single-crystal study, magnetic measurement and wrote most of the paper with O.S.. M.L. and S.K. synthesized the sample and grew single crystals. K.G. performed pyroelectric measurements. Y.H., G.A., H.O., A.S., and W.X. performed spectroscopic studies. M.L.B., M.S.H., and P.B. contributed to X-ray absorption spectroscopy. Y.S., T.A., S.-Q.W., and K.Y. conducted theoretical calculations. H.-Z.K. contributed to the single-crystal analysis. All authors discussed the results and commented on the paper.

## Competing interests

The authors declare no competing interests.
