## [Peer Review File · Nature Communications]

Editorial Note: This manuscript has been previously reviewed at another journal that is not operating a transparent peer review scheme. This document only contains reviewer comments and rebuttal letters for versions considered at *Nature Communications*. Mentions of the other journal have been redacted.

REVIEWERS' COMMENTS:

Reviewer #1 (Remarks to the Author):

This manuscript by Sato and coworkers report on the observation of a macroscopic polarization change in a cobalt dioxolene system which results in a pyroelectric current. The polarization results from the valence tautomeric transition (i.e. intermolecular charge transfer) occurring in polar crystals. I had previously referred this paper [REDACTED] and found it worth of acceptance after corrections and clarifications on a few relevant points. These have been addressed in this revised version.

In detail:

- they clarified the description of the intermolecular interactions, which are deemed crucial to explain the different VT transition temperatures for the two (crystallographically independent) enantiomers and performed a restricted optimization of H coordinate by comparing the electronic density distribution in the two enantiomers;

- they changed the title and reorganized the manuscript as suggested both by me and the other reviewers, improved the reference section, and refined the description of some technical points thus improving the clarity of the manuscript for the non-specialist;

In addition, they performed further characterization at a synchrotron facility (temperature dependent XAS in TEY mode at Co L_{2,3}-edge) trying to elucidate the cause of the temperature mismatch between the transition temperature of the first interconverting molecule and the corresponding peak in the pyroelectric coefficient as required by Ref. #2. Unfortunately, the results on this point were not of much use due to the relevant radiation damage observed on the sample and thus an explanation on this point is deferred to a further studies. I agree with authors that understanding of this discrepancy may well require additional characterization which is beyond the scope of this paper, and does not invalidate the conclusions they reached.

I am then completely satisfied with the responses received, and I think the paper can be accepted for publication in *Nature Communications*. Indeed, as required by the journal guidelines to the reviewers, it is technically sound and provides strong evidence for its conclusions. Further, the observation of pyroelectricity in valence tautomeric compounds and the possibility of fast inducing the polarization change by using light irradiation are of much relevance for researchers in molecular electronics and molecular magnetism, and will trigger further studies in related molecular systems.

Reviewer #2 (Remarks to the Author):

The manuscript by Osamu Sato, Shinji Kanegawa and co-workers is the first report of pyroelectricity accompanying a valence tautomeric transition, exploiting the crystallisation of a cobalt-dioxolene complex in a polar space group. As such, it introduces an additional physical quantity (polarisation) to work with in the research area of switchable molecular materials, which can be combined with the others physical properties changing during the VT transition (molecular volume, molar susceptibility, among the others) to prepare new and multifunctional molecular materials. Even if we feel that its scientific relevance is somehow limited by the directional electron transfer phenomenon previously reported by some of the authors (as reported in ref. 22 in the main text), we believe that the thorough characterisation of the phenomenon, the improvements the paper underwent during the revision process and the potential consequences it may have on the field of switchable molecular materials make it suitable to be published [REDACTED]

Reviewer #3 (Remarks to the Author):

The authors have properly addressed my earlier concerns. Moreover, I feel that Nature Communications is a very suited outlet for this interesting manuscript. As such, I recommend acceptance of the work.